# Rigorous Co-Registration of KOMPSAT-3 Multispectral and Panchromatic Images for Pan-Sharpening Image Fusion

**DOI:** 10.3390/s20072100

**Published:** 2020-04-08

**Authors:** Changno Lee, Jaehong Oh

**Affiliations:** 1Department of Civil Engineering, Seoul National University of Science and Technology, Seoul 01811, Korea; changno@seoultech.ac.kr; 2Department of Civil Engineering, Korea Maritime and Ocean University, Busan 49112, Korea

**Keywords:** KOMPSAT-3, AEISS, co-registration, pan-sharpening, image fusion

## Abstract

KOMPSAT-3, a Korean earth observing satellite, provides the panchromatic (PAN) band and four multispectral (MS) bands. They can be fused to obtain a pan-sharpened image of higher resolution in both the spectral and spatial domain, which is more informative and interpretative for visual inspection. In KOMPSAT-3 Advanced Earth Imaging Sensor System (AEISS) uni-focal camera system, the precise sensor alignment is a prerequisite for the fusion of MS and PAN images because MS and PAN Charge-Coupled Device (CCD) sensors are installed with certain offsets. In addition, exterior effects associated with the ephemeris and terrain elevation lead to the geometric discrepancy between MS and PAN images. Therefore, we propose a rigorous co-registration of KOMPSAT-3 MS and PAN images based on physical sensor modeling. We evaluated the impacts of CCD line offsets, ephemeris, and terrain elevation on the difference in image coordinates. The analysis enables precise co-registration modeling between MS and PAN images. An experiment with KOMPSAT-3 images produced negligible geometric discrepancy between MS and PAN images.

## 1. Introduction

KOMPSAT-3, a Korean earth observing satellite, provides four multispectral (MS) bands (i.e., blue, green, red, near infrared) and one panchromatic (PAN) band. The image fusion or the pan-sharpening combines MS and PAN images into a single image that increase the spatial resolution while simultaneously preserving the spectral information in the MS images. This leads to high spectral resolution and high spatial resolution for various applications, such as visual inspection to identify the textures and shape of the various objects, feature extraction, and map updating. A number of pan-sharpening techniques have been studied, including intensity-hue-saturation (IHS), high-pass filtering (HPF), principal component analysis (PCA), Brovey and wavelet transforms, and Contourlet transform [1]. 

Co-registration is the most critical pre-processing requirement for the fusion of the MS and PAN images, and its accuracy is an important parameter of the fusion product quality [2]. Because MS and PAN CCD (the charge-coupled device) lines are placed with certain offsets, each CCD line captures the same ground target at different times. Under the influence of satellite with the ephemeris and terrain effects remaining, the MS and PAN images cannot be precisely co-registered to each other by a simple image shift. The inaccurate co-registration also impacts further applications like data fusion, change detection, and spectral-signature-based classification.

Conventional image co-registration has been studied out in two ways. First, the image-matching approach performs sub-pixel matching between MS and PAN images, providing tie points that enable geometric modeling between the images [3,4]. In addition, some studies proposed methods consisting of coarse and fine steps [5,6]. Secondly, the sensor modeling approach establishes the geometric relationship between MS and PAN to estimate the geometric discrepancy [7,8]. The aforementioned approaches have advantages and disadvantages. Image-matching methods are usually based on features, which makes it easier to be implemented than sensor modeling. Thus, they are chosen for post-processing when the ephemeris data or the sensor modeling data are not available. There are also many tools available for image registration processes [9,10,11]. However, the performance is highly dependent on terrain features, showing a limited capability over a monotonous terrain (e.g., highly forested area). The sensor modeling approach tends to show higher accuracy, though the rigorous physical modeling process is more complicated and requires the calibration of both the camera and ephemeris models. Currently, the Korea Aerospace Research Institute (KARI) uses image-matching approaches for pan-sharpening image fusion to show mismatch lower than half a pixel in Root Mean Square Error (RMSE) [12], and seeks a rigorous method to reduce the mismatch.

In this study, we investigate the KOMPSAT-3 CCD alignment, attitude change effects, and terrain elevation effects for the improved co-registration of both MS and PAN CCD images. The MS and PAN CCD lines in the KOMPSAT-3 camera system capture the same ground targets at different times due to CCD line offsets. Therefore, even with accurately estimated sensor alignment parameters, exterior effects associated with ephemeris and terrain features still remain. The offsets of CCD lines due to different view directions and terrain elevation variations can lead to mismatches between MS and PAN images.

We implemented a grid of conjugate points to analyze the mismatch patterns between MS and PAN. A grid of points on MS were generated and projected into PAN for conjugate points. The coordinate differences between the MS and PAN points come from the differences in the sensor alignment, ephemeris effects, and terrain elevation. These effects can be modeled sequentially. First, the differences of conjugate point coordinates are modelled for constant image offsets and linear coordinates differences. The model can compensate for the coordinate differences, which are mainly from the CCD line offsets between MS and PAN. Second, the coordinate differences from satellite attitude changes are modelled because the attitude changes over time of PAN and MS data acquisition can lead to different PAN and MS image coordinates. Third, the impact of terrain elevation variation on the row coordinates’ differences is modelled to estimate the row coordinates differences through ground-projection-based sensor modeling between PAN and MS image. Finally, the PAN and MS images are resampled for pan-sharpening, taking account of all the aforementioned effects.

The paper is structured as follows. In Section 2, the KOMPSAT-3 AEISS sensor structure and CCD line offsets are presented, including sensor modeling for image projections and conjugate points computation. In Section 3, the modeling for our rigorous co-registration of both MS and PAN images and its experimental results are presented with respect to CCD line offsets, ephemeris effects, and terrain elevation variations. The conclusion is presented in Section 4.

## 2. KOMPSAT-3 Camera

### 2.1. AEISS Sensor

Figure 1a shows the configuration of KOMPSAT-3 AEISS camera. Blue, PAN1, PAN2, green, red, and near-infrared (NIR) channels are aligned in a unifocal camera of focal length, 8.56215983 meters. Figure 1b depicts the design of PAN and MS CCD alignments. The gaps between sensors in the focal plane correspond to the differences in the projection centers [13]. The camera system is designed to provide a PAN resolution of 0.7 m and an MS resolution of 2.8 m, as presented in Table 1. The pixel sizes of PAN and MS are 8.75 and 35 microns, respectively.

### 2.2. Physical Sensor Modeling

KOMPSAT-3 AEISS is a pushbroom sensor based on a non-linear projection form from a given ground point in an earth-centered, earth-fixed (ECEF) coordinate frame to a point in the body coordinate frame, as shown in Equations (1) and (2) [13]. The ephemeris data are provided from KARI, and they are interpolated for the exterior orientation parameters (EOPs) for a given time. Figure 2 depicts the MS and PAN CCD lines acquiring the same ground target with an offset in the corresponding projection centers. This produces row coordinate differences in MS and PAN images (approximate equation equal to the corresponding ground distance of the offset/spatial resolution in KOMPSAT-3). Note that the amount of row coordinate difference depends on the different view directions.
(1)$$\left[\begin{array}{c} -y^{\left(p\right)}\\ -x^{\left(p\right)}\\ focal \end{array}\right]=\frac{1}{k^{\left(p\right)}}M_{ECI}^{Body}{}^{\left(p\right)}M_{ECEF}^{ECI}{}^{\left(p\right)}\left[\begin{array}{c} X-X_{S}{}^{\left(p\right)}\\ Y-Y_{S}{}^{\left(p\right)}\\ Z-Z_{S}{}^{\left(p\right)} \end{array}\right],$$
(2)$$\left[\begin{array}{c} -y^{\left(m\right)}\\ -x^{\left(m\right)}\\ focal \end{array}\right]=\frac{1}{k^{\left(m\right)}}M_{ECI}^{Body}{}^{\left(m\right)}M_{ECEF}^{ECI}{}^{\left(m\right)}\left[\begin{array}{c} X-X_{S}{}^{\left(m\right)}\\ Y-Y_{S}{}^{\left(m\right)}\\ Z-Z_{S}{}^{\left(m\right)} \end{array}\right],$$
where the upper cases (*p*) and (*m*) denote PAN CCD and MS CCD, respectively. ${\left[\begin{array}{ccc} X & Y & Z \end{array}\right]}^{T}$ are the ground points in the ECEF coordinate frame. ${\left[\begin{array}{ccc} X_{S} & Y_{S} & Z_{S} \end{array}\right]}^{T}$ are the satellite positions in the ECEF coordinate frame. $M_{ECEF}^{ECI}$ is the time-dependent rotation matrix from the ECEF coordinate frame to the earth-centered inertial coordinate frame (ECI). $M_{ECI}^{Body}$ is the time-dependent rotation matrix from the ECI coordinate frame to the body coordinate frame. x,y are the coordinates in the body coordinate frame (y is the scan direction along the image line, and x is the direction along the image column). focal is the focal length and k is the scale factor.

The body coordinates can be converted from the image coordinates using the CCD line alignment information parameters in Equation (3), which is a second-order polynomial. The variable is the column coordinate in pixels, because the KOMPSAT-3 AEISS sensor is a pushbroom sensor of single CCD line vertical to the flight direction. The parameters are determined based on the precise camera calibration before the launch as well as the in-flight calibration with a number of ground control points [14,15,16].
(3)$$\begin{array}{l} x=a_{0}+a_{1}\times samp+a_{2}\times samp^{2}\\ y=b_{0}+b_{1}\times samp+b_{2}\times samp^{2} \end{array},$$
where $a_{0}\sim a_{2},b_{0}\sim b_{2}$ are CCD line alignment parameters, and samp is the column (sample) coordinate in pixels.

### 2.3. Rigorous Co-Registration of MS and PAN Images

Figure 3 shows the flowchart of the rigorous MS and PAN image co-registration in this study. From PAN and MS images, a grid of conjugate points is generated. Note that the conjugate points are not generated using image matching, but their coordinates are computed based on sensor modeling. Given a flat terrain, iterative physical sensor modeling is carried out between PAN and each of the four MS images. For example, ***p’*** on MS in Figure 2 is projected onto the ground surface and then projected into PAN for ***p”***. Similarly, grid points on MS can be projected into PAN for conjugate points. The coordinate differences between the points on MS and PAN are due to the differences in the sensor alignment, ephemeris effects, and terrain elevation. These effects can be modeled as follows.

First, the differences in conjugate point coordinates are modelled to estimate the constant image offsets and linear coordinate differences. This model compensates for the coordinate differences mainly from CCD offsets between MS and PAN. Second, the coordinate differences from satellite attitude changes are analyzed and compensated. Because satellite attitude changes with PAN and MS data acquisition time, the effect of the attitude should be modeled. Third, the effect of terrain elevation variation on the difference in row coordinates is investigated and modelled. We use the sensor modeling via the ground projection between PAN and MS images. The terrain elevation variation introduces coordinate differences that are fixed. Finally, image resampling for the co-registration between PAN and MS is carried out for pan-sharpening, considering all the aforementioned effects.

## 3. Results

### 3.1. Data

The experiment was carried out for Ulaanbaatar, Mongolia with a terrain variation from 1250 to 1950 m. The KOMPSAT-3 image was acquired in 17 July 2016 during 5.17 s with roll, pitch, and yaw angles of −7.7°, −20.0°, and −0.4°, respectively. The satellite azimuth and incidence angles are 145.9° and 23.9°. Table 2 shows the CCD line alignment parameters of each of the four CCD lines, where slightly different values are observed along the scan direction (difference in $b_{0}\sim b_{2}$).

### 3.2. The Effect of CCC Line Offset on the Image Co-Registration

First, a grid of conjugate points was generated in MS and PAN image spaces. Each PAN image coordinate was projected to the ground using a backward model of Equation (1) and then projected to each of the four MS image coordinates using a forward model of Equation (2). For the image projection, the precisely calibrated inner orientation parameters and EOP parameters from the ephemeris were used. Mean elevation flat terrain from shuttle radar topography mission (SRTM) was initially assumed for the iterative projection, because the terrain elevation effect is considered later. Figure 4a shows the grid points in MS and PAN with an interval of 250 pixels. Note that PAN image space is shown in MS image scale. Figure 4b depicts the projection of the image center points in MS onto PAN for conjugate points, where large differences in row coordinate were observed.

We computed the discrepancy of conjugate points between MS and PAN images, as plotted in Figure 5. Note that the difference is computed by subtracting PAN from MS coordinates. In Figure 5a, BLUE CCD shows that row coordinate difference (i.e., scan direction offset) varies by about 17.1–18.8 pixels with a parabolic shape on the x-axis of the column. Figure 5b–d represents that GREEN, RED, and Near IR cases also show similar patterns in the graph. Interestingly, BLUE is the only CCD placed upper than PAN CCD as seen in Figure 1b. On the right side in Figure 5, the differences in column coordinates are observable up to 15 pixels in straight line shapes while the amounts vary with different CCDs.

First, we derived CCD line offsets from row coordinates difference in the first image column as shown in Table 3. The offset ranges from 19 pixels in BLUE to −28 pixels in Near IR. The remaining coordinate differences between MS and PAN CCD lines were modelled using Equation (4). The row and column coordinate differences are modelled using second and first order polynomials, respectively. Table 3 shows the estimated transformation parameters for each of the CCD lines.
(4)$$\begin{array}{l} x^{\left(m\right)}=A\times x^{\left(p\right)}+B\\ y^{\left(m\right)}=C\times {\left(x^{\left(p\right)}\right)}^{2}+D\times x^{\left(p\right)}+E \end{array},$$
where A,B,C,D,E are transformation parameters for CCD offsets.

Figure 6 and Table 4 show the residual after applying the offsets and transformation parameters to each of the four MS images. Most coordinate differences are less than one-pixel level. This can be further improved because this study plans to achieve a difference of less than 0.1 pixels.

### 3.3. The Effect of Attitude Changes on the Image Co-Registration

Previous forward and backward projections are carried out based on the assumptions of accurate ephemeris and flat terrain. First, we analyzed the errors from attitude changes between two acquisition times. The attitude changes in the same image were computed with respect to the change in row coordinates. For BLUE CCD in Figure 7a, both roll and pitch angles changes range from −0.3 to +0.3 in pixels and yaw angle changes vary from −0.08 to +0.04 in pixels. For GREEN CCD in Figure 7b, different amounts of changes are observable. Note that RED and Near IR CCD cases are omitted here because they show the similar patterns to GREEN CCD. Table 5 presents the statistics of the attitude changes in pixels.

We also investigated the effect of satellite attitudes on the MS and PAN image co-registration. We plotted the coordinate differences in the selected grid points with respect to the changes in roll, pitch, and yaw angles in Figure 8. The limit of angle variations were derived from the maximum and minimum attitude angles for the entire image. A linear pattern between the column coordinates’ differences was observed with the change in roll angle. In addition, the row coordinates’ difference was correlated with pitch angle. Yaw angle was associated with both row and column differences, but its effect was smaller than the other angles. It is noteworthy that the roll angle change in the pushbroom sensor was orthogonal to the scan direction, whereas pitch and yaw angles were different from the scan direction viewing angles. The standard deviations in the coordinate differences for each of the four MS sensors are presented in Table 6.

From the analysis, we formed the compensation equation, Equation (5) for attitude changes between two image acquisition epochs derived from MS and PAN CCD line offsets. In Equation (5), the column coordinate correction was linearly modelled using roll and yaw angles. The row coordinate correction is modelled using pitch and yaw angles. Table 7 shows the estimated parameters of the compensation model for attitude changes for each of the four MS bands
(5)$$\begin{array}{l} \Delta column=\alpha \times \Delta roll+\beta \times \Delta yaw\\ \Delta row=\gamma \times \Delta pitch+\eta \times \Delta yaw \end{array}$$
where Δcolumn,Δrow are the differences of column and row coordinates between sub-images, respectively. α,β,γ,η are correction parameters. Δroll,Δpitch,Δyaw are the viewing direction changes in roll, pitch, yaw angles caused by MS-PAN CCD line offsets.

We applied this equation for the differences in the row and column coordinates plotted in Figure 9. The differences become less than our acceptable level, 0.1 pixels. The statistics of each of the four MS sensors are presented in Table 8.

### 3.4. The Effect of Terrain Elevation Variation on the Image Co-Registration

We assumed a flat target terrain derived from SRTM elevation data in the previous section. Here, we tested the ground elevation change in the image projections with some grid points in MS and PAN image spaces. As the red dots show in Figure 10, elevation changes led to less than 2 pixels of row coordinate mismatches. It was also reported that 2.8 km of terrain elevation variation creates about one pixel of mismatch in Quickbird image [17] and the accuracy of DEM is not very critical for determining the relative mismatch between the bands [18]. We can identify that row coordinate differences are highly correlated with terrain elevation variation. Specifically, high row coordinate differences are observed in both RED and Near IR bands.

Therefore it is required to compensate the terrain elevation effect using Equation (6), where $H_{\min},\Delta row_{H_{\min}},$
$H_{\max},\Delta row_{H_{\max}}$ can be derived from a linear regression in Figure 10.
(6)$$\Delta row_{H}=\Delta row_{H_{\min}}+(H-H_{\min})\times \frac{\left(\Delta row_{H_{\max}}-\Delta row_{H_{\min}}\right)}{\left(H_{\max}-H_{\min}\right)},$$
where $\Delta row_{H_{\min}},\Delta row_{H_{\max}}$ are the row coordinate differences at the minimum and maximum terrain heights $H_{\min}$ and $H_{\max}$, respectively. $\Delta row_{H}$ is the row coordinate correction value at the terrain height H.

The terrain elevation compensation parameters for the test data were estimated using the minimum and maximum elevation, 785 and 2789 m ($H_{\min}$ and $H_{\max}$), derived from SRTM data. The blue dots in Figure 10 show the coordinates compensation derived from Equation (6). Table 9 shows the estimations that are different for each of the MS CCD. Note that the Near IR shows the largest row coordinate difference. We applied the estimated compensation parameters and obtained row coordinate differences smaller than 0.1 pixels between PAN and MS images, as shown in Figure 11.

### 3.5. Resampling

We estimated the parameters of the coordinate differences between PAN and each of the four MS sensors due to the initial CCD offsets, attitude changes, and terrain elevation variations. The estimated parameters were used for the co-registration image resampling of four MS images into a PAN image plane for the pan-sharpening. Figure 12 show the result of pan-sharpening in false color before and after the image co-registration. 

### 3.6. Comparison to Image Matching-Based Approach

We compared the rigorous method to image-matching-based co-registrations. For image-matching, Harris operator and Normalized Cross Correlation (NCC) were used for key points’ extraction and image-matching between PAN and MS, as the extracted key points are presented in Figure 13. The result shows that it is not easy to extract points over mountainous areas such as the upper image region. The threshold for NCC matching was set to 0.8 and a total of 750 points were matched after outlier removal. Half of the extracted points were used for co-registration modeling and the other half were used for the co-registration accuracy check. We applied two co-registration modelings: the image shift only and the interpolation [19]. Table 10 shows the accuracy results of co-registration at the check points. The shift-only was not able to achieve sub-pixel accuracy. The interpolation and rigorous methods could achieve sub-pixel accuracy, while the rigorous method show slightly better results. It is notable that the rigorous method may not seem accurate, as the above rigorous analysis showed, but it is likely that the check points are not error-free because they were also extracted using image matching.

## 4. Conclusions

We studied the effects of CCD offsets, attitude effects, and terrain elevation variation on the KOMPSAT-3 MS and PAN image co-registration. The KOMPSAT-3 CCD lines acquire the same ground features with certain offsets in the corresponding projection centers. Therefore, we compensated for the offsets based on the rigorous physical sensor modeling. The remaining mismatches were corrected using the first- and second-order equations formed from the coordinate difference analysis. It is noteworthy that the attitudes between each CCD data acquisition change when imaging the same target are modelled with a linear combination of roll, pitch and yaw angle changes for the time gap. Row coordinate differences between MS and PAN images from terrain elevation variation can be modelled using a linear compensation equation. Finally, we compensated for all aforementioned effects on the KOMPSAT-3 MS and PAN image co-registrations with negligible discrepancy, less than 0.1 pixels. A rigorous co-registration approach is more robust and useful with available ephemeris data, whereas image matching-based co-registrations are less reliable over a monotonous terrains such as desert and forest. In further studies, more co-registration methods based on image matching can be analyzed to compare physical sensor model approaches. 

## Figures and Tables

**Figure 1 sensors-20-02100-f001:**
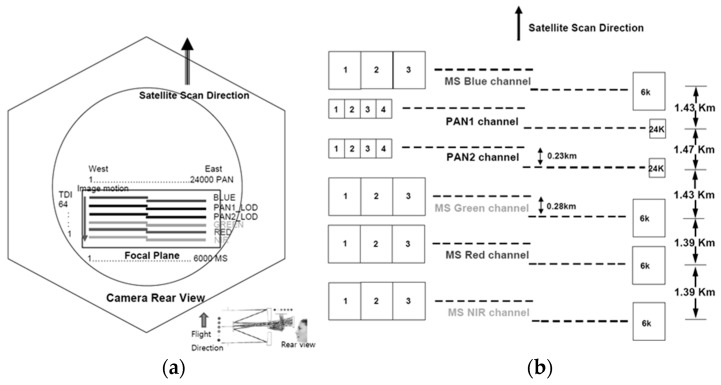
Configuration of KOMPSAT-3 Advanced Earth Imaging Sensor System (AEISS) sensor for (**a**) camera rear view and (**b**) Charge-Coupled Device (CCD) array [13].

**Figure 2 sensors-20-02100-f002:**
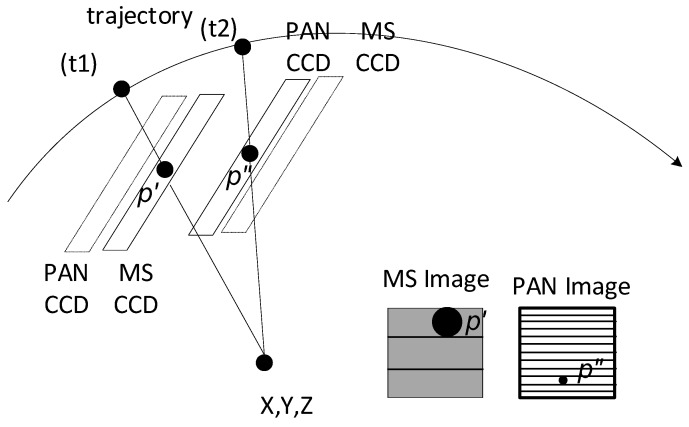
MS and PAN capture the same target in the images at different times because of CCD offsets.

**Figure 3 sensors-20-02100-f003:**
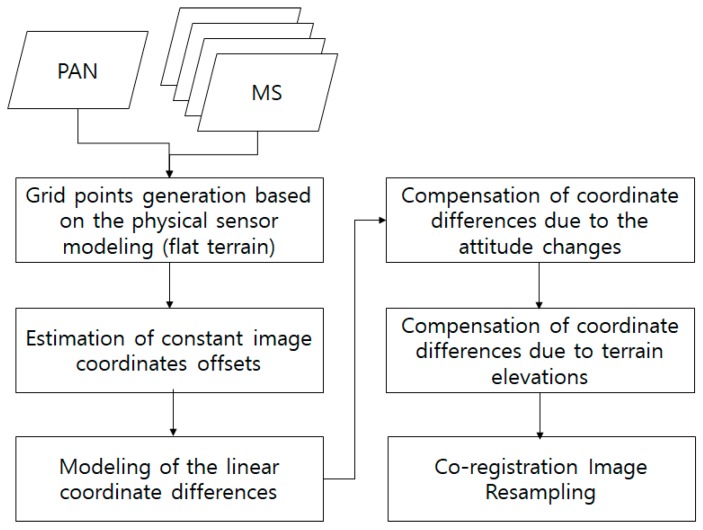
Flowchart of rigorous multispectral (MS) and panchromatic (PAN) image co-registration in this study.

**Figure 4 sensors-20-02100-f004:**
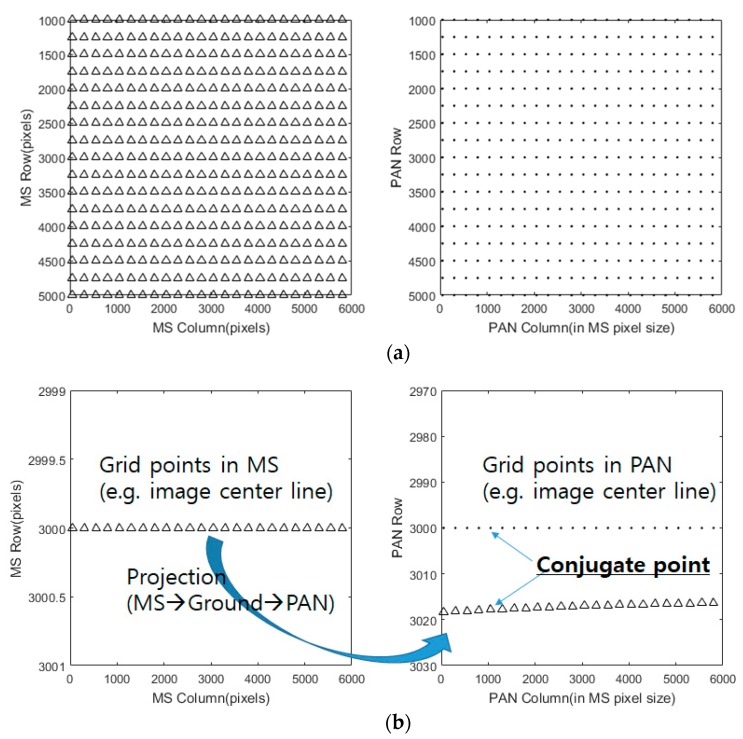
Grid points on MS and PAN (**a**) All grid points with interval 250 pixels in MS and PAN images and (**b**) the MS image center points and the corresponding PAN conjugate points via projection.

**Figure 5 sensors-20-02100-f005:**
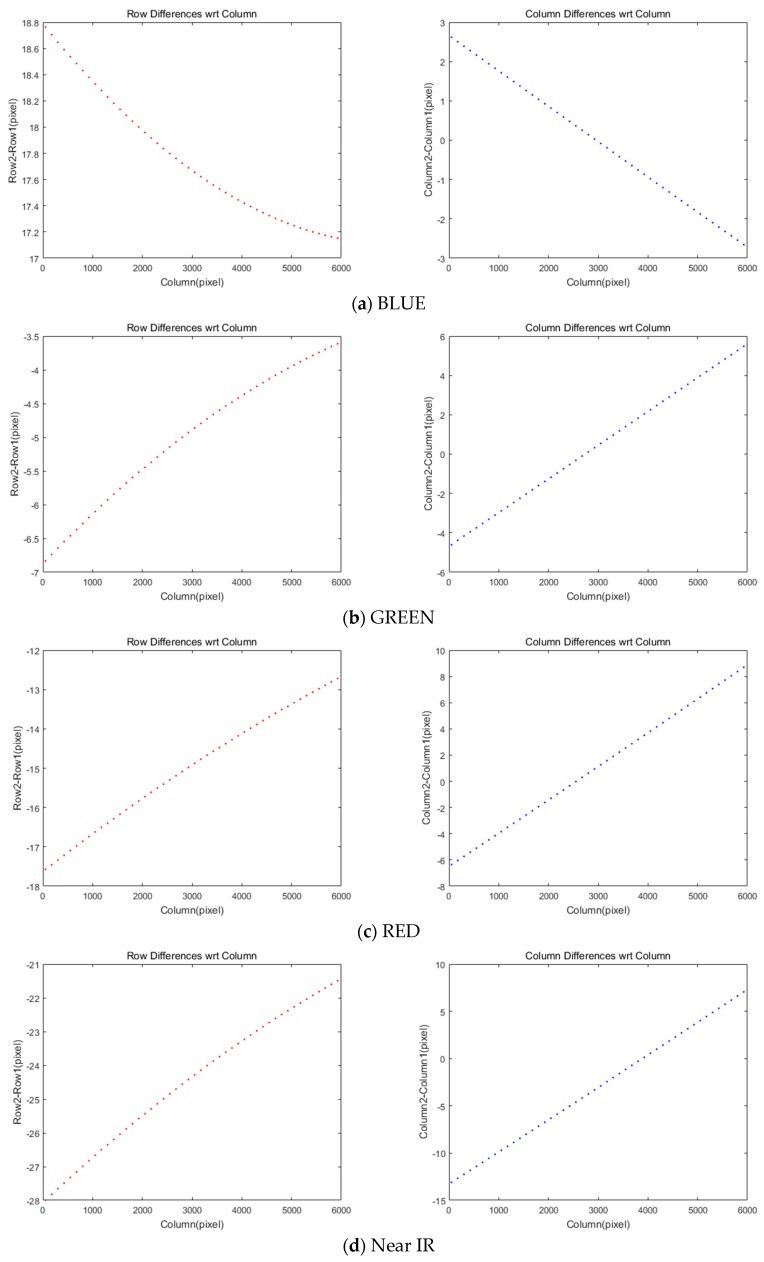
Differences of rows and columns in coordinates between MS and PAN images.

**Figure 6 sensors-20-02100-f006:**
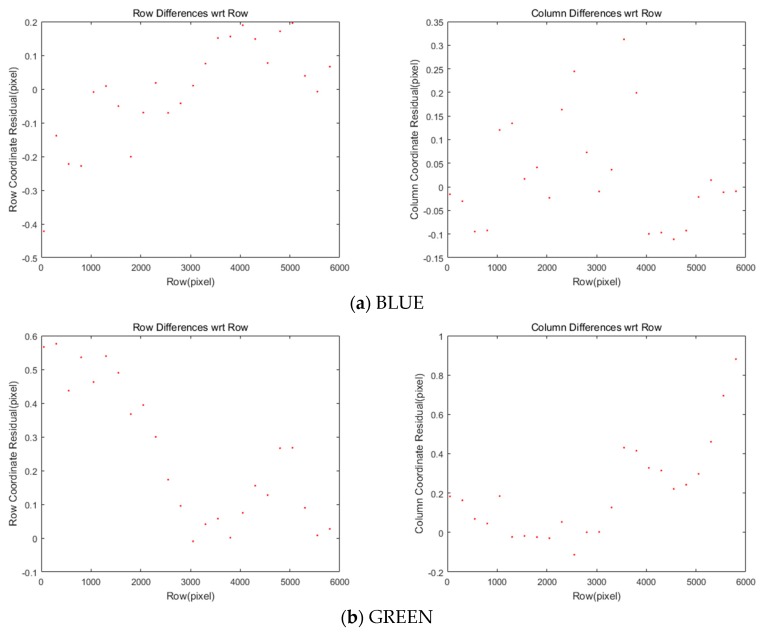
Differences of coordinates between MS and PAN images after CCD offset compensation.

**Figure 7 sensors-20-02100-f007:**
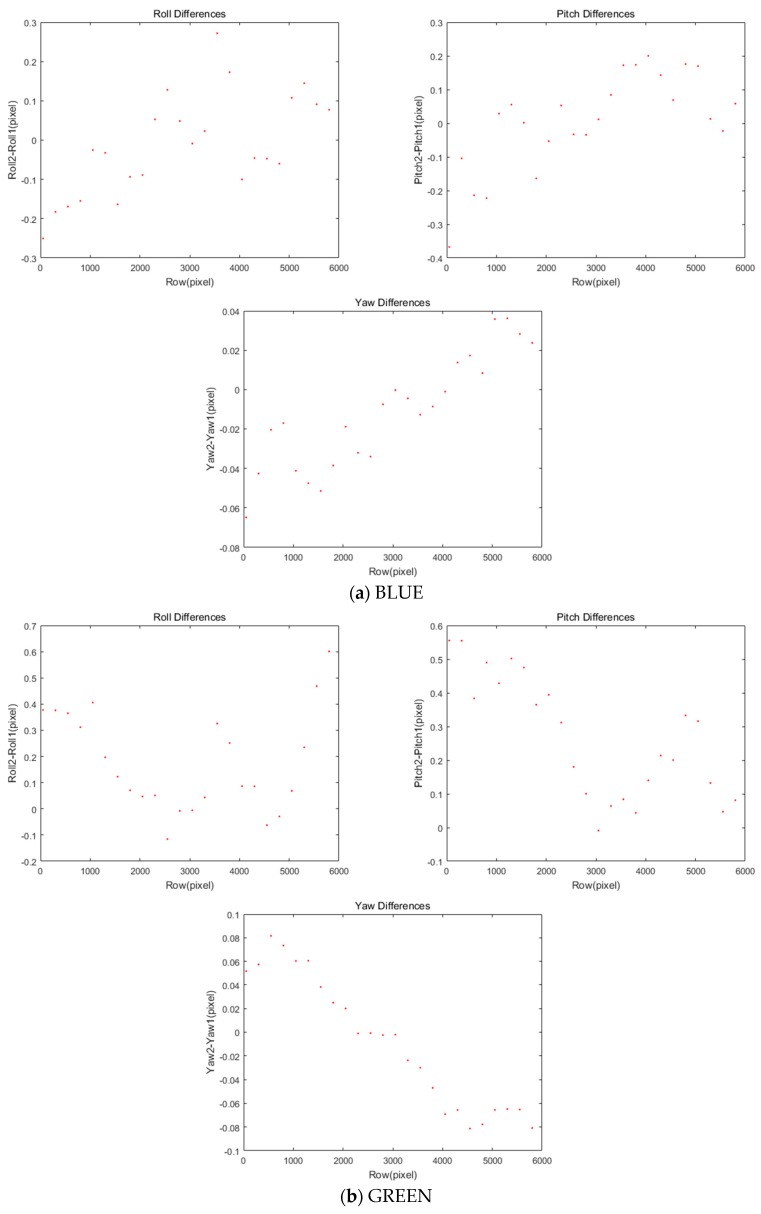
Attitude changes in KOMPSAT-2 in row coordinates.

**Figure 8 sensors-20-02100-f008:**
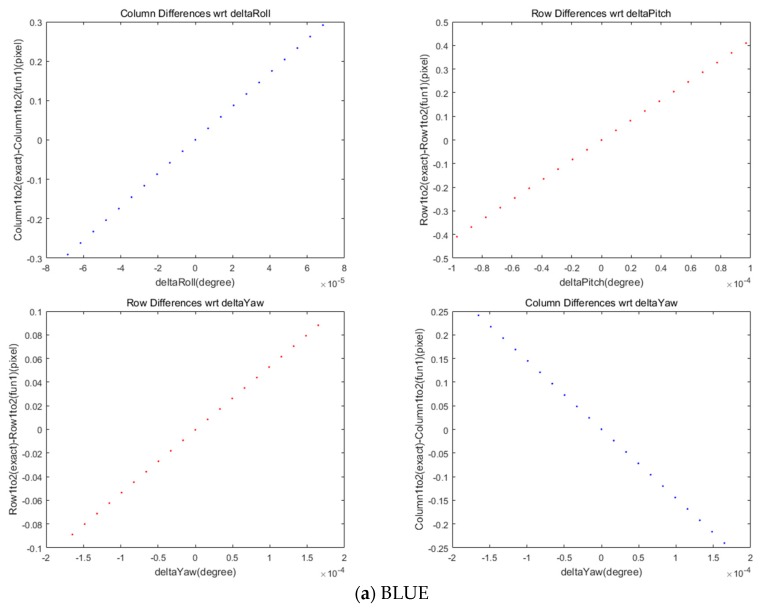
Differences in rows and columns in coordinates between MS and PAN images for attitude changes.

**Figure 9 sensors-20-02100-f009:**
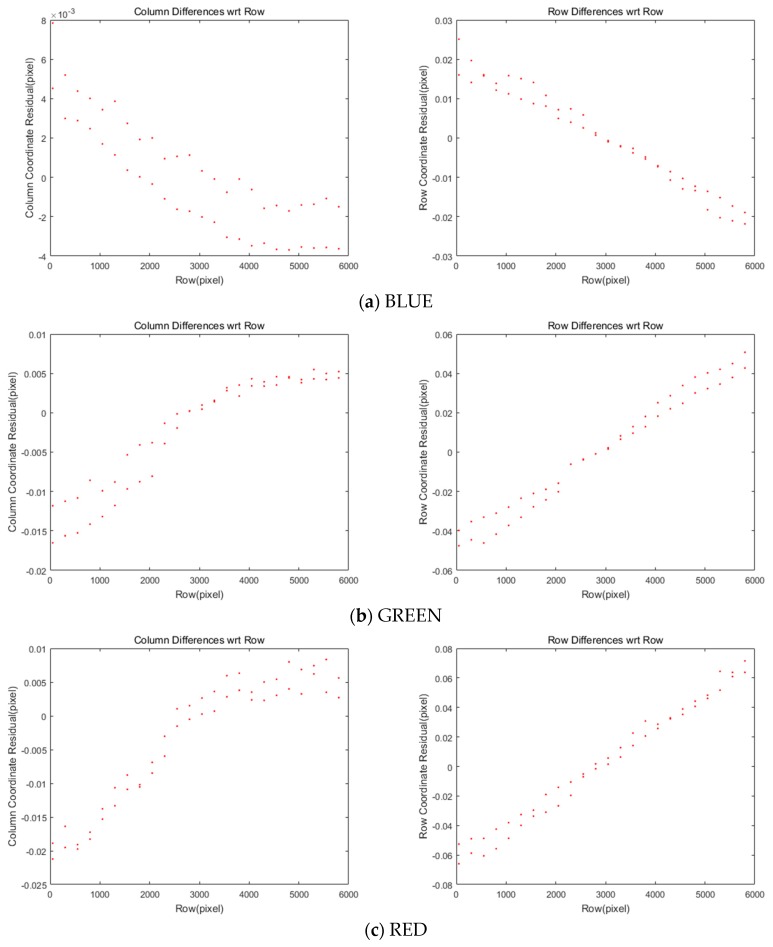
Differences in rows and columns in coordinates between MS and PAN images after attitude change compensation.

**Figure 10 sensors-20-02100-f010:**
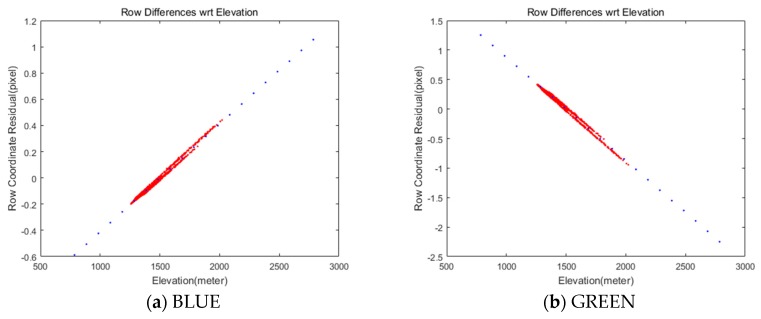
Relationship between row coordinate differences between MS and PAN images due to ground terrain elevation changes (red: row coordinate mismatches, blue: coordinates’ compensation).

**Figure 11 sensors-20-02100-f011:**
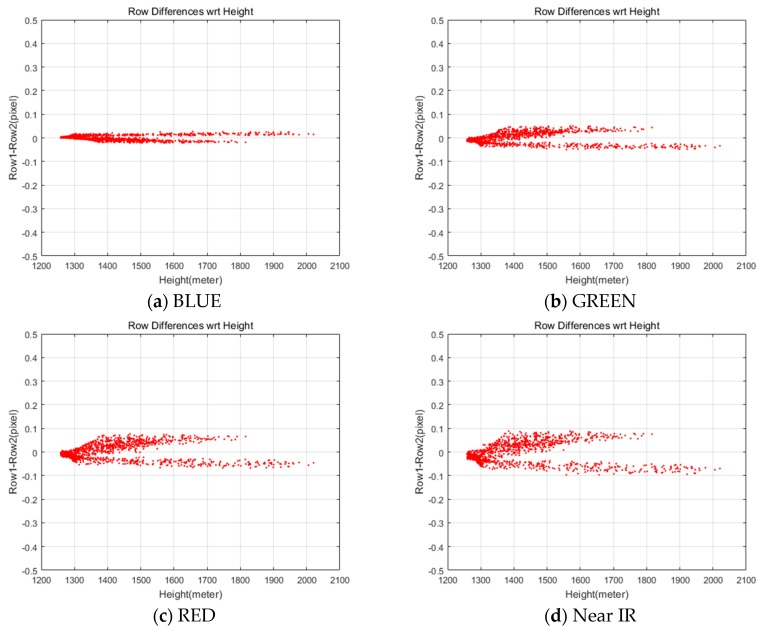
Differences in row coordinate between MS and PAN images after the terrain elevation compensation.

**Figure 12 sensors-20-02100-f012:**
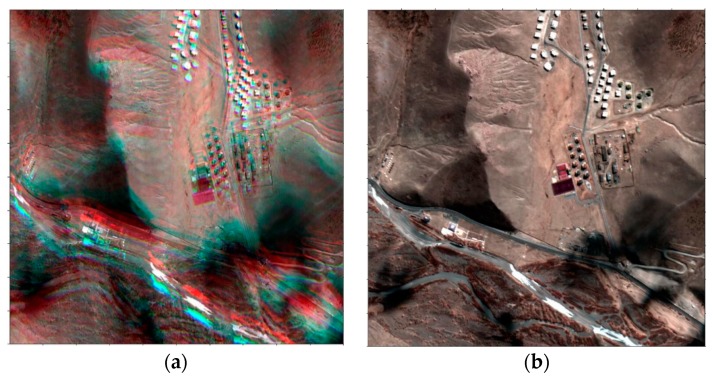
Examples of pan-sharpened images in false color (**a**,**c**) before and (**b**,**d**) after applying image co-registration.

**Figure 13 sensors-20-02100-f013:**
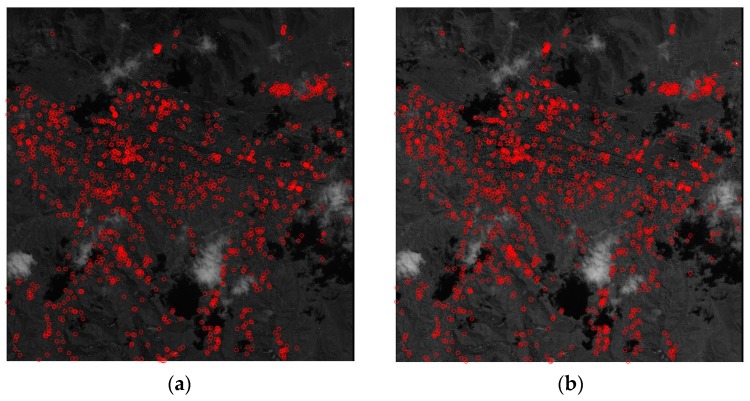
Extracted key points (**a**) PAN, (**b**) MS.

**Table 1 sensors-20-02100-t001:** KOMPSAT-3 specifications.

	PAN (Panchromatic)	MS (Multispectral)
Spectral Bands	450–900 µm	Blue: 450–520 µm Green: 520–600 µm Red: 630–690 µm NIR (Near infra-red): 760–900 µm
Ground Sample Distance (GSD)	0.7 m at nadir	2.8 m at nadir
Focal Length	8.6 m	8.6 m
Swath Width at Nadir	15 km	15 km
Data Quantization	14 bit	14 bit
Charge-Coupled Device (CCD) Detector	Array of 24,000 pixels (2 × 12,000)	Arrays of 4 (RGB and IR) × 6000 pixels (2 × 3000)
Pixel Pitch	8.75 µm	35 µm

**Table 2 sensors-20-02100-t002:** Parameters of CCD line alignment.

CCD	$a_{0}\sim a_{2}$	$b_{0}\sim b_{2}$
PAN	−0.1056, 8.7788 × 10^−^^6^, −9.9221 × 10^−^^14^	5.0044 × 10^−^^6^, −3.8291 × 10^−^^10^, 1.5299 × 10^−^^15^
Blue	−0.1056, 3.5117 × 10^−^^5^, −1.6241 × 10^−^^12^	−0.0174, −8.0087 × 10^−^^9^, 1.2149 × 10^−^^12^
Green	−0.1056, 3.5119 × 10^−^^5^, −1.6298 × 10^−^^12^	0.0373, 4.1031 × 10^−^^9^, −1.3648 × 10^−^^12^
Red	−0.1057, 3.5120 × 10^−^^5^, −1.8738 × 10^−^^12^	0.0547, 1.2393 × 10^−^^9^, −9.5071 × 10^−^^13^
NIR	−0.1055, 3.5119 × 10^−^^5^, −1.8856 × 10^−^^12^	0.0722, 5.2512 × 10^−^^9^, −1.6862 × 10^−^^12^

**Table 3 sensors-20-02100-t003:** Estimated parameters of the coordinate differences due to the initial CCD offsets.

CCD	Initial Offset [pixels]	A	B	C	D	E
Blue	19	0.9992	−0.0000	0.0019	−0.0003	−0.0175
Green	−7	1.0018	−0.0000	−0.0022	0.0004	0.0373
Red	−18	1.0027	0.0000	−0.0015	0.0007	0.0548
NIR	−28	1.0035	0.0000	−0.0027	0.0010	0.0723

**Table 4 sensors-20-02100-t004:** Differences of coordinates between MS and PAN images after CCD offset compensation.

Mean ± Std	Row [pixels]	Col [pixels]
Blue	−0.006 ± 0.154	0.027 ± 0.116
Green	0.252 ± 0.206	0.205 ± 0.243
Red	0.279 ± 0.272	0.158 ± 0.419
NIR	0.433 ± 0.318	0.229 ± 0.592

**Table 5 sensors-20-02100-t005:** Attitude changes in KOMPSAT-2 in row coordinates.

Mean ± Std	Roll (pixels)	Pitch (pixels)	Yaw (pixels)
Blue	−0.013 ± 0.129	0.008 ± 0.144	−0.012 ± 0.029
Green	0.178 ± 0.190	0.266 ± 0.180	−0.009 ± 0.055
Red	0.166 ± 0.280	0.283 ± 0.231	0.001 ± 0.081
NIR	0.308 ± 0.353	0.415 ± 0.284	0.021 ± 0.105

**Table 6 sensors-20-02100-t006:** Standard deviations in the coordinate differences between MS and PAN images for attitude changes.

Std.	Roll→Col	Pitch→Row	Yaw →Row	Yaw→Col
Blue	0.181	0.254	0.055	0.150
Green	0.521	0.375	0.083	0.222
Red	0.562	0.505	0.109	0.291
NIR	0.927	0.714	0.145	0.389

**Table 7 sensors-20-02100-t007:** Estimated parameters of the coordinate differences due to attitude changes.

CCD	α	β	γ	η
Blue	4256.41	−1461.75	4225.50	537.32
Green	4256.24	−1436.37	4205.92	537.53
Red	4256.25	−1428.30	4184.50	537.58
NIR	4256.33	−1420.42	4192.99	525.72

**Table 8 sensors-20-02100-t008:** Differences in rows and columns in coordinates between MS and PAN images after attitude change compensation.

Mean ± Std	Row [pixels]	Col [pixels]
Blue	0.000 ± 0.013	0.000 ± 0.003
Green	0.000 ± 0.029	−0.002 ± 0.007
Red	0.002 ± 0.040	−0.003 ± 0.009
NIR	−0.001 ± 0.052	−0.003 ± 0.013

**Table 9 sensors-20-02100-t009:** Estimated parameters of the row coordinate differences due to terrain elevation variations.

CCD	$\Delta row_{H_{\min}},\Delta row_{H_{\max}}$
Blue	−0.59, 1.05
Green	1.25, −2.25
Red	1.83, −3.30
Nearinfrared (NIR)	2.42, −4.34

**Table 10 sensors-20-02100-t010:** Co-registration root mean square error (RMSE) for the check points.

Method	Row/Col/Total (pixels)
Matching-based (shift)	2.36/0.07/2.36
Matching-based (interpolation)	0.81/0.07/0.81
Rigorous method	0.40/0.28/0.49

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
