# Peer review of "Rigorous Co-Registration of KOMPSAT-3 Multispectral and Panchromatic Images for Pan-Sharpening Image Fusion"

_sensors, 2020, doi:10.3390/s20072100_

Round 1
Reviewer 1 Report
This paper developed a rigorious co-registration algorithm for the low spatial resolution multispectral images and high spatial resolution panchromatic images of Kompsat-3 AEISS. This algorithm would be useful for researchers who are interest in using this new high spatial resolution data sources. I think this paper is worth for publication. However, I have some concerns and hope the authors could revise it before I suggest it for accept.
- Page 4, Line 109, the authors said a grid of tie points is generated. How were those points generated? What about the spatial distribution? How about the accuracy for those tie points? As the tie point is the core for this algorithm, I suggest the authors give a detail description and a distribution map or zoom in map for those tie points would be good.
- All of the experiment and the results are based on one image from Mongolia. Figure 11 only give a very small area where the terrain effects also cannot clearly be identified. I suggest the authors use more images or more case areas to demonstrate the effectiveness of the co-registration method.
- The spatial resolution of SRTM is 30m, but the spatial resolution for MS and PAN image are 2.8m and 0.7m. How to deal with the mismatch in spatial resolution for DEM data and Kompat-3 images? As we can image one SRTM pixel can cover nearly 100 Kompsat-3 MS pixels, those micro-relief is difficult to be depicted from SRTM.
- The size of many sub-figures are different for most of the figures in this paper.
- The information content from Figure 5 to Figure 8 is very limited. The authors could add more information such as standard deviation and confidence interval.
- What’s the meaning of red dots and blue dots in Figure 9?
Author Response
Thank you for the review and suggestions. They helped a lot to improve the manuscript.

Reviewer 2 Report
Please see the attached report.

Author Response

(The authors gave the same response as above.)

Reviewer 3 Report
In this paper, the authors proposed a co-registration approach for multispectral (MS) images and multispectral (MS) images based on the physical sensor modeling for Kompsat-3. The proposed approach has been validated on Kompsat-3 images.
Generally, the proposed idea is very interesting, however, some revisions have to be made and some parts of the manuscript not complete to claim the advantage of the proposed models.
- Could the authors explain what is the main contribution of the proposed approach than over existing methods?
- Could the authors add more details about the generation of a grid of tie-points?
- Could the authors add more clarifications about the flowchart for the rigorous MS and PAN image co-registration?
- The English and format of this manuscript should be checked very carefully.
Author Response

(The authors gave the same response as above.)

Round 2
Reviewer 1 Report
I checked the revised manuscript carefully and very pleased to see the authors accept most of my suggestions. I think the caption of Figure 10 should contain some details of the red dots. After this, I would suggest accept for publication.
Author Response
Thank you for the review and suggestions. They helped a lot to improve the manuscript.
The caption has been updated.
“Figure 10. Relationship between row coordinate differences between MS and PAN images due to ground terrain elevation changes (red: row coordinate mismatches, blue: coordinates compensation).”
Reviewer 2 Report
Although the authors have made significant changes, I still have some concerns.
- In Section 3.5, I suggest that the authors should add the results of using the default registration method in KOMPSAT-3.
- Also, in Section 3.5, the authors should include some conventional registration results. If more time is needed, the authors should request more time from the Editor.
- Regarding my original comment #2, Ref [d] and [e] should be commented in the revised paper because the registration method in those two references consists of two steps: coarse and fine. This is more accurate and very different from conventional methods.
Author Response
Enclosed please find the doc file

Reviewer 3 Report
The authors have revised the manuscript carefully according to my questions. I have no further questions about this manuscript. It could be accepted.
Author Response
Thank you for the review and suggestions. They helped a lot to improve the manuscript
Round 3
Reviewer 2 Report
The paper is acceptable now.